# Heavy Metals in Unprocessed or Minimally Processed Foods Consumed by Humans Worldwide: A Scoping Review

**DOI:** 10.3390/ijerph19148651

**Published:** 2022-07-16

**Authors:** Sonia Collado-López, Larissa Betanzos-Robledo, Martha María Téllez-Rojo, Héctor Lamadrid-Figueroa, Moisés Reyes, Camilo Ríos, Alejandra Cantoral

**Affiliations:** 1School of Public Health, National Institute of Public Health, Cuernavaca 62100, Mexico; sonia.collado@insp.edu.mx; 2Center for Nutrition and Health Research, National Institute of Public Health, Cuernavaca 62100, Mexico; investigador36@insp.mx (L.B.-R.); mmtellez@insp.mx (M.M.T.-R.); 3Department of Perinatal Health, Center for Population Health Research, National Institute of Public Health, Cuernavaca 62100, Mexico; hlamadrid@insp.mx; 4Economics Department and GEOLab-IBERO, Universidad Iberoamericana, Mexico City 01219, Mexico; abraham.reyes@ibero.mx; 5Neurochemistry Department, National Institute of Neurology and Neurosurgery, México City 14269, Mexico; crios@correo.xoc.uam.mx; 6Health Department, Universidad Iberoamericana, México City 01219, Mexico

**Keywords:** heavy metals, food contamination, unprocessed or minimally processed foods

## Abstract

Heavy metals (HM) can be accumulated along the food chain; their presence in food is a global concern for human health because some of them are toxic even at low concentrations. Unprocessed or minimally processed foods are good sources of different nutrients, so their safety and quality composition should be guaranteed in the most natural form that is obtained for human consumption. The objective of this scoping review (ScR) is to summarize the existing evidence about the presence of HM content (arsenic (As), lead (Pb), cadmium (Cd), mercury (Hg), methylmercury (MeHg), and aluminum (Al)) in unprocessed or minimally processed foods for human consumption worldwide during the period of 2011–2020. As a second objective, we identified reported HM values in food with respect to Food and Agriculture Organization of the United Nations (FAO) and the World Health Organization (WHO) International Food Standards for Maximum Limits (MLs) for contaminants in food. This ScR was conducted in accordance with the Joanna Briggs Institute (JBI) methodology and PRISMA Extension for Scoping Reviews (PRISMA-ScR); advance searches were performed in PubMed, ScienceDirect and FAO AGRIS (Agricultural Science and Technology Information) databases by two reviewers who independently performed literature searches with specific eligibility criteria. We classified individual foods in food groups and subgroups according to the Global Individual Information Food Consumption Data Tool (FAO/WHO GIFT). We homologated all the reported HM units to parts per million (ppm) to determine the weighted mean HM concentration per country and food group/subgroup of the articles included. Then, we compared HM concentration findings with FAO/WHO MLs. Finally, we used a Geographic Information System (GIS) to present our findings. Using our search strategy, we included 152 articles. Asia was the continent with the highest number of publications (*n* = 79, 51.3%), with China being the country with the largest number of studies (*n* = 34). Fish and shellfish (*n* = 58), followed by vegetables (*n* = 39) and cereals (*n* = 38), were the food groups studied the most. Fish (*n* = 42), rice (*n* = 33), and leafy (*n* = 28) and fruiting vegetables (*n* = 29) were the most studied food subgroups. With respect to the HM of interest, Cd was the most analyzed, followed by Pb, As, Hg and Al. Finally, we found that many of the HM concentrations reported exceeded the FAO/OMS MLs established for Cd, Pb and As globally in all food groups, mainly in vegetables, followed by the roots and tubers, and cereals food groups. Our study highlights the presence of HM in the most natural forms of food around the world, in concentrations that, in fact, exceed the MLs, which affects food safety and could represent a human health risk. In countries with regulations on these topics, a monitoring system is recommended to evaluate and monitor compliance with national standards. For countries without a regulation system, it is recommended to adopt international guidelines, such as those of FAO, and implement a monitoring system that supervises national compliance. In both cases, the information must be disseminated to the population to create social awareness. This is especially important to protect the population from the consumption of internal production and for the international markets of the globalized world.

## 1. Introduction

Food contamination with heavy metals (HM) is a global concern for human health [1], as HM are non-biodegradable pollutants that can accumulate and migrate into soil environments [2]. The ingestion of food with these pollutants represents one of the most important human exposure routes to these metals [2,3]. HM are electropositive elements found in all ecosystems; their natural concentrations vary accordingly to the local geology, and human activities redistribute them in areas not naturally enriched with them [4]. While certain HM, such as nickel (Ni), iron (Fe), magnesium (Mn), copper (Cu) and zinc (Zn), in foodstuffs at low concentrations are essential components for biological and physiological functions and metabolic processes, including cytochrome and enzyme functions [3,5,6], others, such as lead (Pb), cadmium (Cd), mercury (Hg), as well as other toxic elements such as arsenic (As) and aluminum (Al) [7], are toxic even at low concentrations, and are classified as non-essential to metabolic and biological functions [6,7,8]. In this study, we refer to all these afore-mentioned toxic elements as HM. Exposure to HM is related to acute and chronic toxic effects since they can disturb major metabolic processes through their ability to accumulate in vital organs (e.g., liver, heart, kidney and brain) [9]. They can substitute essential elements (e.g., calcium can be substituted by lead), or set up a biological and antioxidant imbalance, in addition to altering hormone, enzyme and the central nervous system functions [9]. Finally, HM exposure induces oxidative stress generation, which may result in different kinds of cancers, neurological disorders and damage to the kidney function as well as endocrine abnormalities [5].

HM toxicity to humans depends on different factors, such as dose, route of exposure, chemical species and some individual characteristics (such as age, gender, and nutritional status) [5]. Additionally, HM are widely distributed in the environment through natural processes, and mainly through anthropogenic activities, such as industrial, domestic and agricultural production and use, and through mining and smelting activities [5]. The presence of these elements in the environment varies depending on natural geographic differences, and past or current contamination [10]. In addition, the high demand for food and population increases have released different pollutants into the environment that contaminate the food chain through agrochemicals, municipal wastewater, raw sewage and industrial effluents [11], or by their entry into the marine and coastal environment as a consequence of coastal pollution [12].

Different from other pollutants, HM are unable to decompose and are non-degradable and they can be accumulated along the food chain, becoming a threat to human health [13]; their exposure may occur through food, as well as through other important ways [1]. HM presence in food varies according to the different routes or sources of contamination, defined as intrinsic and extrinsic factors. Intrinsic factors include the seasons, soil, water, atmospheric deposits, animal feeding regimen [14] and volcanic and vehicle emissions [15]. Additionally, there are extrinsic factors that contribute to food contamination, such as food technological processes, packaging, transportation, storage [14] and culinary procedures tools and cooking methods [16,17]. In this way, the levels of HM in foods are different, according to the type of food, environment and agricultural and industrial food process [10].

Natural or unprocessed foods refer to the edible parts of animals (muscle, offal, eggs or milk), plants (seeds, fruit, leaves, stems, tubers and roots) fungi and algae, succeeding just after their separation from their nature status, and minimally processed foods refers to natural foods that were altered by methods and processes to preserve them [18]. These food groups provide different essential nutrients to human organisms, e.g., many plant source foods provide different micronutrients and bioactive compounds and animal food sources provide amino acids, vitamins and minerals [18]. Unprocessed and minimally processed foods are important components of the human diet because of their high nutritional value [12,19]. In this ScR, we are interested in studying the presence of HM in the most natural form of the foods, in their raw stages, or the ones with minimally processed methods designed to preserve natural foods and that could reflect environmental sources of contaminants. Different from natural foods that could be contaminated by intrinsic factors, processed or ultra-processed foods and cooked foods could include chemical contaminants (as additives) or process contaminants (added during the cooking, heating or storage process of food at home or industries or during their transportation) that have an important impact on food quality and safety and may represent different routes or sources of contamination [14,20]. The presence of HM in foods has public health implications; food safety is necessary to maintain food quality [21] and, therefore, some agencies, at the global level, have established standards for Maximum Limits (MLs) of metals and other toxic elements in food. We highlight the Codex Alimentarius Commission, managed by the Food and Agriculture Organization of the United Nations (FAO) and the World Health Organization (WHO), which through the establishment of MLs of contaminants, such as HM, promotes the safety and quality food production around the world [22,23].

The main objective of the present Scoping Review (ScR) is to summarize the existing evidence about the presence of specific HM (As, Pb, Cd, Hg, MeHg and Al) contents in unprocessed or minimally processed foods for human consumption worldwide. As a second objective, we identified how those reported HM values in food were found with respect to FAO/WHO International Food Standards for MLs for contaminants in food.

## 2. Methods

This ScR was conducted following the Joanna Briggs Institute (JBI) methodology for conducting Scoping Reviews [24] and the PRISMA Extension for Scoping Reviews (PRISMA-ScR) [25]. Before any search was conducted, the protocol was registered in Open Science Framework in April 2021 [26] (https://osf.io/25gps, accessed on 29 April 2021). 

This ScR was based on PRISMA key elements, Population, Concept and Context (PCC), and considered as participants the studies that reported information about HM content in foods. The concepts investigated were HM and unprocessed or minimally processed foods. Regarding context, worldwide information from countries, cities or localities in areas with or without identified environmental pollution was considered.

This study included original research studies published from 1 January 2011 to 31 December 2020, in English or Spanish. We considered cross-sectional and longitudinal studies and published reports that measured the presence of the following HM As, Pb, Cd, Hg, MeHg and/or Al, in unprocessed or minimally processed foods (cereals, roots and tubers, pulses, seeds and nuts, milk, eggs, fish and shellfish, meat, vegetables and fruits) in any part of the world.

### 2.1. Search Strategy and Data Extraction

The literature searches were conducted in duplicate. In the first step, two researchers performed independent literature advanced searches on the PubMed, ScienceDirect and FAO AGRIS databases between 25 May and 25 October 2021. The search terms were selected according to MeSH Terms from two main clusters: Heavy Metals and Food Contamination. Terms were combined within each cluster using “OR”; these clusters were then combined using “AND”. The string word “NOT” term was added to exclude for systematic reviews and for water. Details of the electronic search strategy and the filters applied in all searches are presented in Table 1. As a second step, we conducted the search in titles, abstracts and full-text articles. After compiling search results, duplicate articles were eliminated. Subsequently, in order to compare the obtained information, we excluded (a) reviews, systematics reviews or meta-analysis studies; as well as studies exploring the presence of the interest HM in: (b) animal feed, (c) water, (d) fungi and plants such as cilantro and parsley, (e) culinary ingredients (substances obtained by industrial processes such as pressing, centrifuging, refining or extracting) such as honey and vegetable oils, and (f) hunted animals; (g) studies without a measure of central tendency and/or without an exact or precise value of HM; (h) studies that test a methodology to validate the content of HM in food or that make an experiment to modify the HM content in foods; (i) studies exploring the HM content in an inedible part of the food; (j) studies without the specification that the studied food was natural, raw or without any cooking process; (k) studies that reported all their HM content information in a way that could not be classified with the methodology used, such as information by food group (if there were individual foods reported, the article was not excluded); and (l) studies without information of the weight base HM analysis (as dry, wet or fresh weight). Figure 1 shows the PRISMA flowchart diagram of the literature review pipeline.

### 2.2. Statistical Analysis and Data Presentation

The identified literature was analyzed according to the extent of worldwide evidence, measured by the number of publications by continent and country, followed by the group and subgroup foods and the HM studied in them and by those articles that reported concentrations of HM exceeding the FAO/WHO MLs for contaminants.

### 2.3. Creation of Variables for the Food Groups and Subgroups 

In order to compare the obtained information, we classified each food item reported according to the Global Individual Information Food Consumption Data Tool (FAO/WHO GIFT) [26], into nine food groups, cereals, roots and tubers, pulses, seeds and nuts, milk, eggs, fish and shellfish, meat, vegetables and fruits, and in their corresponding subgroup; more information about the individual food items that compose each food group and subgroup is presented in Appendix A. We classified food items within each food group considering the FAO/WHO GIFT Tool and the Codex Alimentarius International Food Standards set by the FAO and the OMS [28]. We also used the AGROVOC Multilingual Thesaurus [29] when food items were not reported on those tools.

### 2.4. Heavy Metals (HM) in Food Geographical Analysis

In addition, for each reviewed study, we estimated the HM mean from the longitudinal studies that reported the information on more than one occasion in the same study area. Additionally, in each study, when the measurements of HM were individually reported from different places within the same country, we estimated the HM mean separating the noncontaminated areas from the contaminated ones. Then, we made the conversion and homologation of all the reported HM units to parts per million (ppm), to finally, construct the weighted mean (WM), defined as:(1)$$X_{j}=\sum_{i=1}^{i=N}\left(\frac{n_{ij}}{N_{j}}x_{ij}\right)$$
where (*i*) identifies the study under consideration, while (*j*) identifies the HM present and (*n*) is the number of food group observations. In this sense, $\left(n_{ij}\right)$ indicates the number of food group observations contained in the study (*i*) with HM (*j*) present. $\left(N_{j}\right)$ represents, for each country and food group, the sum of the number of observations (*n*) for the present HM (*j*). $Nj=\sum_{i=1}^{N}n_{ij}$, where ***N*** is the number of studies considered for each country and food group. (*x_ij_*) represent, for each country and food group, the quantity of HM (*j*) present in the study considered (*i*). Therefore, *X_i_* represents, for each country and food group, the amount of HM (j) present, weighted by the number of observations contained in the studies in that country. The weighted mean for each HM was thus calculated at the country level and according to each food group.

Finally, we created maps of the evidence; therefore, the information of the 152 included studies were represented in QGIS open-source Geographic Information System (GIS) [30] (more information is presented in Appendix A). The HM ppm weighted mean concentration data were classified according to the natural breaks methodology; this method uses a computing algorithm in order to minimize differences between the data values in the same class and maximize the differences between classes [31].

### 2.5. Reference Values for Maximum Limits (MLs) of Heavy Metals (HM) in Foods

We compared the subgroup mean reported value of HM in foods per study with those HM MLs reported at the Codex Alimentarius International Food Standards FAO/WHO [22].

## 3. Results

### 3.1. Study Inclusion

Using our search strategy, the database search resulted in 2733 studies. After the removal of duplicated studies (*n* = 105), we obtained 2628 unique studies screened for title and abstract. Then, we considered 359 full-text studies for reading. Of these, 207 were excluded according to the following exclusion criteria: processed food (5), not natural experiment (15), publication before 2011 (2), hunted animals (6), reported animal feed (1), without a measure of central tendency and/or without an exact or precise value of HM (33), studied other metals (12), without weight base HM analysis (57), culinary or plant ingredients (2), fungi studies (33), with sample size ≤ 1 or without sample size (24), food items with a cooking process or not reported as raw or natural food (3), studies of foods that were identified ≤3 times per food group (1), not identified as edible food (1), all results presented as food group (6), evaluated HM thorough storage containers (5), and does not report the country of origin (1). Finally, 152 studies were included (Figure 1).

### 3.2. Worldwide Evidence

Worldwide evidence distribution by country of study and their corresponding continent are presented in Figure 2, Map 1. Studies were conducted in 43 countries; in Asia, the studies were conducted mainly in 13 countries, with China being the country with the highest number of studies (*n* = 34), followed by Iran (21 studies). In Europe, studies focused on 16 countries, where Italy was the country with the highest number of publications (*n* = 14 studies). In America, studies were identified in six countries, and Brazil had the highest number of studies (*n* = 7). In Africa, evidence was found in seven countries, with Nigeria (*n* = 3) being the predominant one, and in Oceania, only two studies from Australia were included. Thus, Asia was the continent with the highest number of publications (*n* = 79, 51.3%), followed by Europe (*n* = 45, 29.9%), America (*n* = 15, 9.7%), Africa (*n* = 11, 7.1%) and Oceania (*n* = 2, 1.3%).

### 3.3. Principal Food Groups and Heavy Metals Identified

Figure 3 shows the distribution of worldwide studies and their corresponding HM concentrations reported in the three principal food groups from which we found more evidence (fish and shellfish, vegetables, and cereals). The fish and shellfish group was the most studied food group (*n* = 58), as Figure 3 Map 2 shows; Italy (*n* = 9) was the country with the highest number of studies reporting on the analysis of HM, followed by China (*n* = 8). The highest weighted average for As levels were found in Germany (mean = 127 ppm), followed by Turkey (mean = 58.3 ppm), while the highest weighted average for Cd in this food group was found in Iran (mean = 1.6 ppm), Bangladesh (mean = 0.96 ppm) and China (mean = 0.68 ppm). Nigeria was the country with the highest levels of Pb reported (mean = 20.15 ppm). The higher weighted average levels of Hg were found in Germany (mean = 248.5 ppm), followed by Iran (mean = 6.79 ppm) and Bangladesh (mean = 5.24 ppm). Canada had the highest weighted average values for MeHg (mean = 0.17 ppm).

Vegetables was the second most studied food group (*n* = 39), with China being the country with the highest number of studies (*n* = 12). The highest weighted average levels of As were found in Portugal (mean = 0.38 ppm), and the highest weighted average for Cd was found in South Africa (mean = 0.99 ppm). In this food group, the highest weighted average for Pb was found in Serbia (mean = 3.35 ppm), and for Hg, it was found in China (mean = 0.038 ppm) (Figure 3 Map 3). Finally, with respect to the cereals food group (*n* = 38), China was also the country with the highest number of studies (*n* = 16) and with the highest Hg weighted average (mean = 0.67 ppm). Bangladesh had the highest weighted average for As (mean = 0.21 ppm), while Australia for Cd (mean = 15.3 ppm) and Pb (mean = 271.87 ppm) (Figure 3 Map 4).

The map distribution of the other less reported food groups is presented in (Appendix A). The roots and tubers group (*n* = 25) was mainly studied in China (*n* = 5). The highest As and Cd weighted average were reported in Iran, Pb in Serbia and Hg in Croatia. Meat (*n* = 20) had the highest number of studies in China (*n* = 2); the highest As and Hg weighted average was found in Kuwait, Cd in South Korea and Pb in Tunisia. With respect to the pulses, seeds and nuts food group (*n* = 16), China (*n* = 6) was the country with most evidence published. The highest levels of As were found in Bangladesh, Cd and Hg in China, and Pb in Iran for this food group. Fruits (*n* = 13) were mainly studied in Bangladesh (*n* = 3), which was the country with the highest weighted average for As, and Germany reported the highest levels of Cd and Pb. Milk (*n* = 12) presented the highest weighted average levels of As and Hg in Bangladesh, Cd in Ethiopia and Pb in South Korea. Finally, the eggs group (*n* = 8) was the least studied.

### 3.4. Food Subgroups by Food Group and Heavy Metals Identified

Table 2 describes the number of food groups and food subgroups reported in the different studies included in this ScR, as well as the number of determinations from each HM evaluated in the corresponding food subgroup. It is essential to highlight that some articles reported various food items (food groups), as well as different HM measurements. We found that, in the fish and shellfish food group, Hg/MeHg followed by Cd were the most studied HM, and fish (*n* = 42), crustaceans (*n* = 17) and mollusks (*n* = 14) were the most studied subgroups. In the cereals food group, rice (*n* = 33) was the most reported food subgroup and Cd was the most studied HM. In the vegetables group, all subgroups were studied, with leafy (*n* = 28) and fruiting (*n* = 29) vegetables being the most reported, and Cd followed by Pb being the most studied HM in these subgroups. Regarding the roots and tubers group, potatoes (*n* = 21) followed by carrots were the most studied subgroups, and Cd and Pb followed by As were the most reported HM. In the pulses, nuts and seeds food group, pulses (*n* = 11) were the most studied, and Cd, Pb and As were the most reported HM. In fruits food group, tropical Fruits (*n* = 7) were the most studied subgroup. Chicken eggs (*n* = 6) and cow milk (*n* = 11) were the most reported food subgroups in the eggs and milk food groups; these groups were the least studied and Cd, Pb, As and Hg/MeHg were measured on them.

### 3.5. Heavy Metals (HM) in Food Subgroups in Comparison with the FAO/OMS Maximum Limits (Mls) for Contaminants in Food

We compared the HM obtained from the studies included in this review with the MLs of HM established by FAO/OMS. Table 3 shows the number of studies per country that reported food subgroups that exceeded MLs (the completed data extracted of each study are presented in Appendix A). Even though Pb and Cd were the HM with more reported information in our search for the different food subgroups, there were many food subgroups without established ML values. Additionally, there are not established ML values for Al and Hg; therefore, our comparison and results are limited to the HM and food subgroups for which MLs have been established. 

With respect to cereals group, in the rice subgroup, China was the country with the highest number of studies reporting an excess of As and Cd. The reported means As exceeding the MLs ranged from 0.24 ppm to 0.78 ppm. The highest concentrations of As were reported by Rahman et al. in Bangladesh (mean = 0.56 ppm and 0.78 ppm) [105]. The reported means exceeding the Cd MLs in rice ranged from 0.41 ppm to 73.0 ppm, with Australia being the country with the highest Cd concentrations [59]. For the maize and wheat subgroups, there are only established ML values for Pb and Cd; the Cd concentrations that exceeded the MLs in maize were in the range of 0.23 ppm to 1.5 ppm, with Iran being the country with the highest Pb concentration [164]. Regarding Cd, the range was from 0.1 to 16.17, and the highest concentration was reported by Peng et al. in China; it is important to highlight that the study samples were collected from a major agricultural area located near an important industrial base [172]. We also found values that exceeded the limits in the wheat subgroup for Pb (mean = 1.85 ppm) and Cd (mean = 0.41 ppm) in Iran [164], and for Pb (mean = 3.61 ppm) and Cd (mean = 0.2) in China, where study samples were taken from amended soils with biogas slurry [172].

In the roots and tubers group, the potato subgroup had the highest number of studies that reported values of Pb and Cd that exceeded MLs, with Pb values from 0.1 ppm to 6.07 ppm. the highest value was reported by Yang et al. in China [76]. The range of Cd values was from 0.11 to 1.09, with this last concentration being reported by Guerra et al. in Brazil. Carrot was another subgroup that had various values reported that exceeded the MLs [154]; for Pb, the range of values that exceeded the MLs ranged from 0.3 ppm to 2.5 ppm, and for Cd from 0.21 ppm to 0.49 ppm, with Servia [150] and Croatia [127] being the countries with the highest reported concentrations, respectively.

Regarding the pulses, seeds and nuts group, the FAO/OMS MLs are available only for the subgroup of pulses and for Pb and Cd. The value that exceeded the MLs for Pb ranged from 0.6 ppm to 95 ppm, with the lentil being the individual food item that had the highest reported value by Pirsaheb et al. in Iran [164]. Regarding Cd, the values that exceeded the MLs ranged from 0.17 ppm to 3.39 ppm, with China having the highest concentration and the sample was obtained from a coal-mining city [126]. With respect to the milk group, MLs values have only been established for Pb; half of the studies reported concentrations that exceeded MLs, with Pb exceeding concentrations from 0.23 ppm to 1.48 ppm and South Korea being the country that reported the highest value [161]. Cow milk was the most studied food subgroup in this category.

Regarding the eggs group, five studies reported Pb concentration results; four of them had levels exceeding the MLs. The values of reported Pb levels that exceeded the MLs ranged from 0.15 ppm to 4.06 ppm; chicken eggs were the most studied subgroup and duck eggs were the food subgroup with the highest concentration reported in Thailand [95].

With respect to the fish and shellfish group, in the fish subgroup, the values of Pb exceeding ML concentrations ranged from 0.35 ppm to 20.15 ppm, and the highest value was reported in Nigeria, whose sample was obtained from effluents from university and fishing activities [51]. Mollusks exceeding MLs were only found in Iran (mean = 11.3 ppm), whose sample originated from an area with anthropogenic polluting activities [62]. Exceeding MeHg MLs for tuna were found in Italy (mean = 1.7 ppm) [98], and in Mexico for shark (mean = 1.65 ppm) [181].

In the meat group, four Asian countries and one African country reported Pb values exceeding MLs in red meat, ranging from 0.1 ppm to 9.2 ppm; the latter value was reported by Kim et al. in South Korea in a sample of domestic pig muscle [161]. Regarding poultry meat, four Asian countries reported Pb values exceeding MLs ranging from 0.17 ppm to 5.2 ppm, the latter value having been found in a sample of 61 chicken muscles in South Korea [161]. Pb levels exceeding the MLs were reported for offal read meat and offal poultry in three [35,56,161] and four countries [41,48,80,161], respectively, ranging from 0.63 ppm to 47.7 ppm for offal red meat and from 0.25 ppm to 37.6 ppm for offal poultry.

There were many countries where HM concentration exceed the MLs for Pb and Cd in the vegetables group; China [76,126,157,172], Iran [110,146], Bangladesh [105], Brazil [147,154], Spain [106], Croatia [127], Portugal [44] Germany [46], Serbia [150], South Africa [37] and Turkey [135] were found exceeding MLs in at least one vegetables subgroup. In the leafy vegetables subgroup, exceeding MLs means values ranged, for Pb, from 0.3 ppm to 12.8 ppm, and for Cd from 0.2 ppm to 21.9 ppm; in the stalk and stem vegetables subgroup, Cd mean concentrations ranged from 0.19 ppm to 9.18 ppm, while in the Brassica subgroup, Pb concentration mean values ranged from 1.0 ppm to 15.6 ppm and for Cd from 0.05 ppm to 18.6 ppm; for the bulb vegetables subgroup, Pb exceeded values ranging from 1.0 ppm to 10.4 ppm and for Cd from 0.05 ppm to 4.47 ppm; for fruiting vegetables, Pb exceeding MLs varied from 0.056 ppm to 10.2 ppm, and for Cd from 0.05 ppm to 15.3 ppm; and finally for legume vegetables, the exceeding MLs for Pb varied from 1.3 ppm to 10.9 ppm, and for Cd from 0.17 ppm to 0.23 ppm. China was the country that reported the highest number of studies with exceeding MLs for Pb and Cd in all the vegetable subgroups, and most of the exceeding foods were sampled from a coal-mining city [76,126]. Only one study from Bangladesh by Rahman et al. reported the highest Cd concentration in the legume vegetable subgroup (mean = 0.23 ppm) [105].

Finally, with respect to the fruits group, the FAO/OMS MLs have been only established for Pb. Tropical fruits was the subgroup with the highest number of studies that reported Pb levels exceeding the MLs, ranging from 0.28 ppm to 0.93 ppm. Bangladesh was the country with the most articles reporting this excess and the country with the highest levels reported by Rahman et al. in guava fruit [105]. Pome fruits was another subgroup whose Pb values exceeded the MLs, ranging from 0.45 ppm to 29.3 ppm. The highest value was reported by Hoffen et al. in Germany, specifically in apples grown in an urban garden close to traffic areas; the same study reported the highest values in stone fruits (mean = 23.2 ppm) and soft fruits (mean = 59.5 ppm) [58].

## 4. Discussion

We systematically summarized the existing evidence of HM content in foods (natural or minimally processed). This represents an important topic because dietary exposure to HM has been recognized as a public health concern (2), posing a serious threat to food safety and human health. Exposure to HM through foods can contribute to their accumulation in the human body, which eventually causes oxidative stress and could lead to different harmful effects [184] and diseases [9], especially in prenatal maternal exposure, where it has been associated with adverse birth outcomes, such as low birth weight [185].

Our review identified 152 studies, published between 2011 and 2020, with most of the study evidence having been reported recently, during the period of 2018–2022 (69 studies). This is consistent with the relevance of the topic, which has been increasingly studied in recent years, and provides evidence of the emerging global concern of the presence of toxic elements in the food we eat. Our findings also show that most of the evidence comes from Asian countries, particularly China, which reflects the lack of evidence in the rest of the world. This lack of evidence is the worst when we evaluated the number of publications by country, as only 43 countries have at least one article published about the topic, representing 22% of the countries around the world. There are some areas in America and Africa where our searches and study inclusion criteria found no evidence or reports regarding the measurement of HM concentration in foods, which should be a concern to monitor for the corresponding agencies.

The amount of evidence in China could be related to the documented information about widespread soil contamination with HM in that country [186,187], but actually, we cannot arrive at conclusions about this gap in the evidence, and the reasons are outside the scope of this study.

Other previous reviews suggest that contaminated zones could impact the presence of HM in food. A review carried out in Bangladesh showed that vegetables in sewage-irrigated areas were heavy-metal- and metalloid-contaminated, and that fish species were highly contaminated with Cd, Pb and Cr [188]. Another review explored different sources of HM contamination in soil–food crop subsystems worldwide and found that, in high-income countries, the deposition of particulate matter and the use of industrial effluents and sewage sludge as fertilizers were the primary contamination sources, while irrigation with inadequately treated effluent or sludge were the principal sources of contamination in low and middle countries [189].

In this review, we found the presence of toxic HM in all food groups around the world, and we identified that the most studied food groups were fish and shellfish, vegetables, and cereals, which is consistent with other worldwide reviews that have documented that aquatic foods, fruits, vegetables and major staple foods, such as tubers, are the major HM hosts [190].

With respect to International MLs, we found that many of the HM concentrations reported exceeded the FAO/OMS MLs established for Cd, Pb and As globally in all food groups, mainly in vegetables, followed by the roots and tubers, and cereals group. The food subgroups with the highest number of studies reporting measures that exceeded the MLs were leafy, brassica and fruiting from the vegetables group, potato and carrot from the roots and tuber group and rice from cereals group. This is consistent with other reviews that identify exceeding MLs for Pb and Cd in spinach, jute mallow and tomato [190]; the concentrations of Pb, Cd and Ni in some fruits were above the recommended values by the European Union in the Eastern Nigeria [191]. Additionally, a review carried out in Bangladesh found that vegetables in sewage-irrigated areas were HM contaminated and that fish species were highly contaminated with Cd, Pb and Cr [188]. A systematic review conducted in Iran about toxic metals in consumed rice brands for human consumption shows that 88% of the rice consumed does not meet the national standard and WHO/FAO guideline requirements [192].

Some of the studies included in this ScR that reported HM concentrations exceeded the FAO/OMS MLs; had samples that obtained from polluted or contaminated areas, such as with Pb and Cd in China [126], whose vegetable food group sample was obtained from a coal-mining city, or with Pb for the mollusks subgroup in Iran, whose sample was obtained from an area with anthropogenic polluting activities [62]. It is important to highlight that the articles included in this review evaluated only food for human consumption, so it is a major concern that areas known to be contaminated are food-producing areas.

Nevertheless, even though we found food subgroups exceeding MLs in contaminated zones, we also found reported values below the detection limits (<LOD) in contaminated zones, such as in Portugal, where cabbage samples were obtained from a mining area [43], or Turkey, where corn samples were obtained from an area whose soil was impacted by industrial and municipal wastewaters [135], or Cd for rice in China, whose sample was collected in a major agricultural area near an important industrial base [141].

Because we studied foods in their most natural form, we expected that concentrations for most of the reported foods were under the LOD values; instead, our results show that there are many subgroups above the MLs. This highlights that, even though we found HM in food from contaminated zones, this is not the only source of contaminants in food, so agricultural zones must study and report the presence of HM in the foods they produce.

Among the strengths of our study, we can affirm that, to the best of our knowledge, this is the first ScR to explore the published evidence regarding HM content in unprocessed or minimally processed foods consumed by humans worldwide. Additionally, we followed a systematic process for mapping the evidence and followed international criteria for the process of this study, following the JBI methodology for conducting ScRs [24] and the PRISMA Extension for ScRs [25]. We used an international food group and subgroup classification system [28], so food information could be comparable with international information and international dietary recommendations. We also compared HM MLs with international references reported at the Codex Alimentarius International Food Standards FAO/WHO [22], so MLs were the same for all countries, which allowed us to use the same criteria for all articles; this comparison highlighted the need to increase the evidence related to this topic in order to establish MLs for other HM, such as Al and Hg, and to establish MLs in all food groups. Our study reflects the countries where we found evidence and reflects the need to study places where there is a lack of information. Such findings point toward a need to monitor and regulate HM in foods for human consumption, especially in those areas that are contaminated and food-producing areas.

The limitations of our literature review include the fact that, because our search results demonstrate high heterogeneity among the methodologists across studies (which represented a challenge in making the comparison, mapping and summarizing the evidence results), we had to follow strict exclusion criteria, which could have left out information that could help us to obtain a more precise panorama of the global situation. Additionally, we grouped our food results accordingly to the FAO GIFT tool, so we could not include studies’ information that was reported by food group; this decreased the number of food items reported per article. We excluded mushrooms and plant foods because of their high heterogeneity on the subspecies and methods of determination, but those were part of the food subgroups we studied, which could make us lose valuable information with respect to the HM present on the vegetables food group. With respect to the HM exceeding MLs, our results are limited to the few available HM MLs for FAO/OMS guidelines; this does not represent that we did not find evidence of exceeding MLs for Hg and Al in all food groups, or As, Pb, or Cd for many of the food subgroups, but rather that there are no established MLs for all HM and all food subgroups studied. Therefore, our results should be interpreted with caution. Our study results do not reflect a country’s problem with the presence of HM in food; they only reflect the zones and periods where HM were found, and the areas where studies have been performed. Another limitation is that studying the causes of the presence of HM was outside of the scope of this study, so we cannot conclude about the impact of natural sources and human activities related to agriculture (such as the use of pesticides or dumping of sewage and industrial wastewater in cultivated land) in the presence of HM in food. Lastly, we identified the study of the causes of the presence of HM in food; the study of HM content in mushrooms, herbs and spices; the study of HM presence in processed foods; and the establishment of As, Pb, Cd, Me, Hg, MeHg and Al MLs for all food groups as opportunities for future research.

## 5. Conclusions

The presence of HM in foods was found around the world and in all food groups studied. This study constitutes a starting point for the importance of exploring the presence of toxic elements in foods. In countries with regulations on these topics, a monitoring system is recommended to evaluate and monitor compliance with national standards. For countries without a regulation system, it is recommended to adopt international guidelines, such as those of the FAO, and implement a monitoring system that supervises national compliance. In both cases, the information must be disseminated to the population to create social awareness. This is especially important to protect the population from the consumption of internal production and for the international markets of the globalized world.

## Figures and Tables

**Figure 1 ijerph-19-08651-f001:**
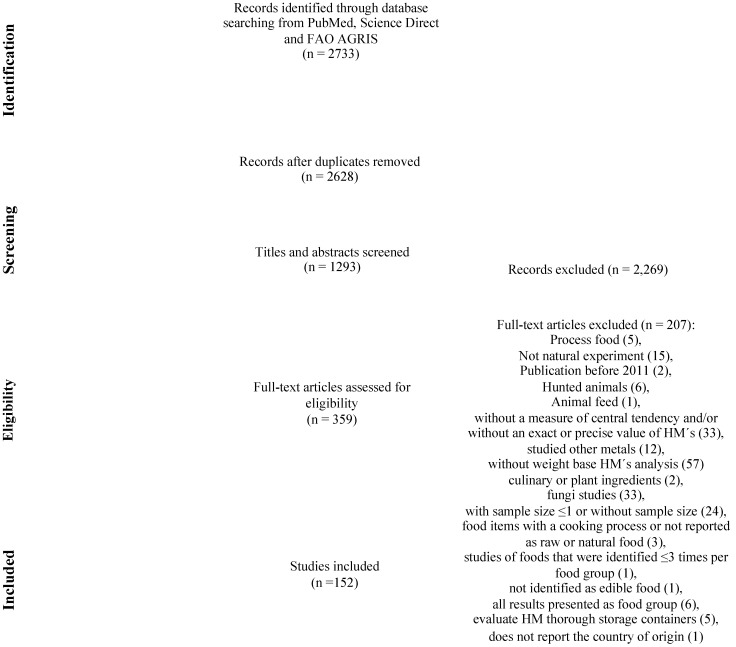
PRISMA flowchart of the results. Adapted from Ref. [27].

**Figure 2 ijerph-19-08651-f002:**
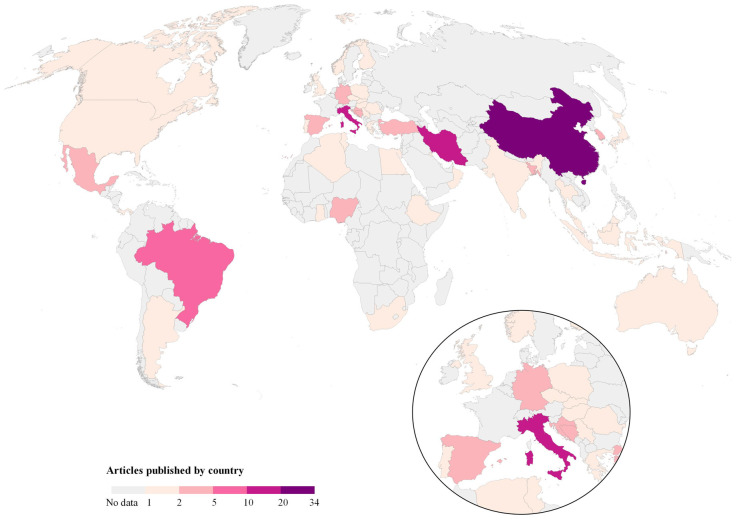
Map 1. Worldwide evidence distribution of the included studies by country. Source: Authors’ own elaboration. Data from Refs. [32,33,34,35,36,37,38,39,40,41,42,43,44,45,46,47,48,49,50,51,52,53,54,55,56,57,58,59,60,61,62,63,64,65,66,67,68,69,70,71,72,73,74,75,76,77,78,79,80,81,82,83,84,85,86,87,88,89,90,91,92,93,94,95,96,97,98,99,100,101,102,103,104,105,106,107,108,109,110,111,112,113,114,115,116,117,118,119,120,121,122,123,124,125,126,127,128,129,130,131,132,133,134,135,136,137,138,139,140,141,142,143,144,145,146,147,148,149,150,151,152,153,154,155,156,157,158,159,160,161,162,163,164,165,166,167,168,169,170,171,172,173,174,175,176,177,178,179,180,181,182,183].

**Figure 3 ijerph-19-08651-f003:**
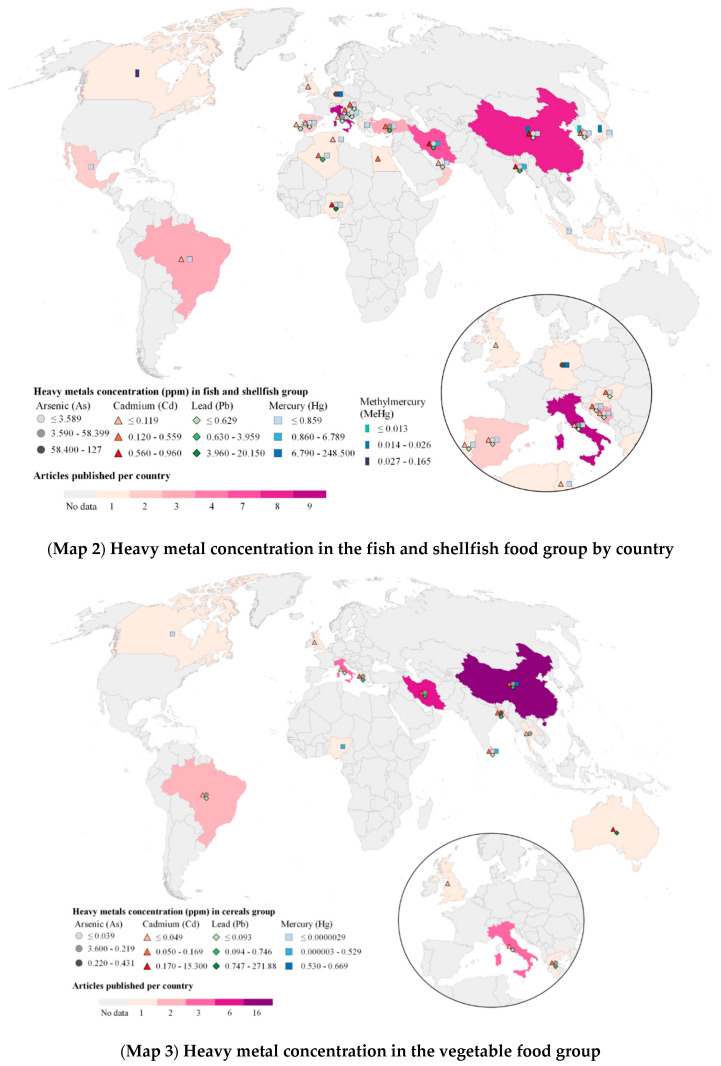
Heavy metals concentration in parts per million units in the most reported food groups worldwide. Source: Authors’ own elaboration. Data from Refs. [32,33,34,35,36,37,38,39,40,41,42,43,44,45,46,47,48,49,50,51,52,53,54,55,56,57,58,59,60,61,62,63,64,65,66,67,68,69,70,71,72,73,74,75,76,77,78,79,80,81,82,83,84,85,86,87,88,89,90,91,92,93,94,95,96,97,98,99,100,101,102,103,104,105,106,107,108,109,110,111,112,113,114,115,116,117,118,119,120,121,122,123,124,125,126,127,128,129,130,131,132,133,134,135,136,137,138,139,140,141,142,143,144,145,146,147,148,149,150,151,152,153,154,155,156,157,158,159,160,161,162,163,164,165,166,167,168,169,170,171,172,173,174,175,176,177,178,179,180,181,182,183].

**Table 1 ijerph-19-08651-t001:** Search strategy in databases.

Database	Search	Results
PubMed #5	(((#1) AND (#2)) NOT (#3)) NOT (#4)	2324
#4	(Review) OR (Review Literature)	1,184,623
#3	(Water)	406,953
#2	((Food contamination) OR (Food Contaminations)) OR (Contamination, Food)	34,956
#1	((Metals, Heavy) OR (Heavy Metals) OR (Heavy Metal))	198,701
Filters applied in every search: Books and Documents, Case Reports, Clinical Study, Clinical Trial, Journal Article, Observational Study, Randomized Controlled Trial, English, Spanish, MEDLINE, from 2011–2020		
ScienceDirect		
#1	(((Metals, Heavy) OR (Heavy Metals)) OR (Heavy Metal)) AND ((Food contamination) OR (Food Contaminations) OR (Contamination, Food)) NOT (Water) NOT ((Review) OR (Review Literature)))	266
Filters applied: Year: 2011–2020 Title, abstract, keywords		
FAO|AGRIS		
#2	(((Metals, Heavy) OR (Heavy Metals)) OR (Heavy Metal)) AND ((Food contamination) OR (Food Contaminations) OR (Contamination, Food)) NOT (Water) NOT ((Review) OR (Review Literature)))	6
#1	(((Metals, Heavy) OR (Heavy Metals)) OR (Heavy Metal)) AND ((Food contamination) OR (Food Contaminations) OR (Contamination, Food)) NOT (Water) NOT ((Review) OR (Review Literature)))	137
Filters applied search #2: language:(Spanish), publication Date: [2011 TO 2020]		
Filters applied search #1: language:(English), publication Date: [2011 TO 2020]		

# Refers to the number of advance searches in the databases.

**Table 2 ijerph-19-08651-t002:** Description of number of food groups and food subgroups, reported in the different studies included in this review, as well as the number of times that heavy metals were evaluated in the corresponding food subgroup.

				Number of Articles by Heavy Metals Studied				
Food Group Studied *	Number of Articles Reviewed by Food Group	Food Subgroup Studied *	Number of Articles that Report the Food Subgroup	As	Pb	Cd	Hg/MeHg	Al
Cereals	37	Maize	6	5	6	6	2	1
		Millet	1	0	0	1	0	0
		Rice	33	12	16	25	8	3
		Wheat	7	2	6	7	1	1
		Others (barley, oat)	1	0	1	1	0	1
Eggs	8	Chicken egg	6	2	3	3	2	1
		Duck egg	1	0	1	1	1	0
		Fish egg	1	0	0	1	0	0
		Turtle egg	1	1	0	1	0	0
Fish and shellfish	58	Cephalopods	8	3	5	5	9	0
		Crustacean	17	7	10	11	14	1
		Fish	42	15	22	26	39	2
		Molluses	14	7	10	10	14	1
		Offal fish	4	3	2	2	4	0
Fruits	13	Citrus fruits	3	0	2	3	0	0
		Pome fruits	5	1	4	5	0	0
		Soft fruits	7	1	6	7	1	1
		Stone fruits	3	1	3	3	0	0
		Tropical fruits	7	3	6	7	0	0
		Watermelons	3	1	3	3	0	0
Meat	20	Offal poultry	6	2	4	5	4	1
		Offal red meat	8	3	5	7	4	0
		Poultry meat	10	5	7	8	5	1
		Red meat	16	8	11	15	8	0
Milk	12	Cow milk	11	4	9	9	5	0
		Ewe milk	2	0	2	2	1	0
		Goat milk	1	0	1	0	0	0
Pulses, seed and nuts	16	Nuts and seeds	4	0	3	4	0	0
		Pulses	11	7	10	11	3	0
		Soybeans	4	2	4	4	1	0
Roots and tubers	25	Beetroot	4	0	3	4	0	0
		Carrot	12	5	8	12	1	1
		Potato	21	6	17	20	2	1
		Radish	5	2	3	5	1	0
		Others	6	1	5	6	0	0
Vegetables	39	Brasica vegetables	21	6	16	19	5	1
		Bulb vegetables	19	4	13	19	2	0
		Fruiting vegetables	29	10	25	26	4	3
		Leafy vegetables	28	9	24	27	7	2
		Legume vegetables	10	2	9	10	0	0
		Stalk and sterm vegetables	9	4	7	9	2	0
			Total of times HM reported	144	292	350	150	22

***** Adapted from FAO and WHO Global Individual Food Consumption data Tool (GIFT). Ref. [28] Abbreviations: As: Arsenic; Pb: Lead; Cd: Cadmium; Hg/MeHg: Mercury and or Methylmercury.

**Table 3 ijerph-19-08651-t003:** Number of articles that reported food subgroups that exceeded the Maximum Limits of heavy metals established by FAO/OMS per country.

Classification FAO/WHO Global Individual Food Consumption Data Tool (GIFT) ^a^		FAO/OMS Maximum Limits (MLs) ^b^				Number of Articles per Country that Exceed Maximum Limits (MLs) in our Study ^c^			
Food Group	Subgroup-Short Name	As	Pb	Cd	MeHg	As	Pb	Cd	MeHg
Cereals	Rice	0.2		0.4		Thailand: 1 Bangladesh: 1 Iran: 2 China: 2		China: 3 Iran: 1 Australia: 1	
	Maize		0.2	0.1			Bangladesh: 1 Greece: 1 Iran: 1 China: 1	Bangladesh: 2 Iran: 1 China: 2	
	Wheat		0.2	0.2			Iran: 1 China: 1	Iran: 1 China: 1 UK: 1	
Roots and tubers	Potato		0.1	0.1			Bangladesh: 1 Brazil: 1 China: 2 Croatia: 1 Germany: 1 Iran: 1 Poland: 1 Serbia: 1 Slovakia: 1	Brazil: 1 China: 2 Croatia: 1 Iran: 1 Poland: 1	
	Other starchy roots and tubers:		0.1	0.1					
	Beetroot						Brazil: 1 Serbia: 1		
	Carrot						Bangladesh: 1 Brazil: 1 China: 1 Croatia: 1 Germany: 1 Poland: 1 Serbia: 1	Croatia: 1 Germany: 1 Poland: 1 Serbia: 1	
	Radish						China: 1 Iran: 1	Iran: 1	
Pulses, seeds and nuts	Pulses		0.1	0.1			Bangladesh: 2 Brazil: 1 Croatia: 1 Iran: 1	Bangladesh: 1 Brazil: 1 China: 1 Iran: 1	
Milk	Milk		0.02				Bangladesh: 2 Ethiopia: 1 Hungary: 1 Mexico: 1 South Korea: 1		
Eggs	Eggs		0.1				Australia: 1 Bangladesh: 1 India: 1 Thailand: 1		
Fish and shellfish	Freshwater, diadromous and marine fish		0.3				Argelia: 1 Bangladesh: 1 Bosnia and Herzegovina: 1 China: 1 Iran: 1 Italy: 1 Nigeria: 1 Turkey: 1		
	Mollusks			2.0				Iran: 1	
	Cephalopods			2.0					
	Tuna				1.2				Italia: 1
	Shark				1.6				Mexico: 1
Meat	Red meat		0.1				Bangladesh: 1 China: 1 Kuwait: 1 Nigeria: 1 South Korea: 1		
	Poultry		0.1				Bangladesh: 1 India: 1 South Korea: 1 Thailand: 1		
	Offal red meat		0.2				Kuwait: 1 Nigeria: 1 South Korea: 1		
	Offal poultry		0.1				Ghana: 1 Malaysia: 1 Tunisia: 1 South Korea: 1		
Vegetables	Leafy vegetables		0.3	0.2			Bangladesh: 1 Brazil: 2 China: 4 Croatia: 1 Germany: 1 Iran: 2 Portugal: 1 Serbia: 1 Spain: 1	Bangladesh: 1 China: 4 Croatia: 1 Germany: 1 Iran: 2 Serbia: 1 South Africa: 1 Spain: 1	
	Stalk and steem vegetables			0.1				China: 2	
	Brassica vegetables		0.1	0.05			Bangladesh: 1 Brazil: 1 China: 6 Croatia: 1 Germany: 1 Serbia: 1	Brazil: 1 China: 6 Croatia: 1 Germany: 1 Serbia: 1 South Africa: 1	
	Bulb vegetables		0.1	0.05			Bangladesh: 1 Brazil: 1 China: 3 Croatia: 1 Iran: 2	Bnagladesh: 1 Brazil: 1 China: 3 Croatia: 1 Ghana: 1 Iran: 3 South Africa: 1	
	Fruiting vegetables		0.05	0.05			Bangladesh: 2 Brazil: 2 China: 4 Germany: 1 Serbia: 1 Turkey: 1	Bangladesh: 3 Brazil: 2 China: 5 Germany: 1 Ghana: 1 Iran: 1 Serbia: 1 South Africa: 1 Turkey: 1	
	Legume vegetables		0.1	0.1			Bangladesh: 1 Brazil: 1 China: 2 Germany: 1 Iran: 1	Bangladesh: 1 China: 1	
Fruits	Tropical fruits		0.1				Bangladesh: 3 Brazil: 1 China: 1		
	Stone fruits		0.1				Bangladesh: 1 Germany: 1		
	Pome fruits		0.1				Bangladesh: 1 Brazil: 1 Germany: 1		
	Soft fruits (berries and other small fruits)		0.1				Brazil: 1 Germany: 1		

^a^ Adapted from the FAO and WHO Global Individual Food Consumption data Tool (GIFT). ^b^ Adapted from the FAO and WHO General Standard for Contaminants and Toxins in Food and Feed. ^c^ According to the studies’ information that are included in this Scoping Review. NOTE: All reported means were converted to ppm (kg/g) to facilitate comparison. There are no established FAO/OMS Maximum Limits (MLs) for mercury (Hg) and aluminum (Al). Abbreviations: As: Arsenic; Pb: Lead; Cd: Cadmium; MeHg: Methylmercury.

## Data Availability

The datasets used and/or analyzed during the current study are available from the corresponding author upon reasonable request.

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
