# Peer review of "Heavy Metals in Unprocessed or Minimally Processed Foods Consumed by Humans Worldwide: A Scoping Review"

_ijerph, 2022, doi:10.3390/ijerph19148651_

Round 1

Reviewer 1 Report

The manuscript entitled ‘Heavy metals in unprocessed or minimally processed foods consumed by humans worldwide: a scoping review’ is within the scope of the journal. It is a well-written manuscript that reveals the level of ‘Heavy Metals’ in processed and minimally processed foods however, the following comments need to be addressed before publication:

General comments:

1.     Kindly check the entire MS for grammatical errors and English language.

2.     What was only unprocessed or minimally processed foods taken as the part of the study?

3.     It seems a kind of bibliometric analysis therefore, kindly provide a bibliometric graph for the study.

Specific comments:

Introduction

It is not clear what are the route/sources of HM contamination in these food materials? Kindly supplement this information.

Conclusion

It should be more specific along with future recommendations.

Author Response

Reviewer #1

Comments and Suggestions for Authors

The manuscript entitled ‘Heavy metals in unprocessed or minimally processed foods consumed by humans worldwide: a scoping review’ is within the scope of the journal. It is a well-written manuscript that reveals the level of ‘Heavy Metals’ in processed and minimally processed foods however, the following comments need to be addressed before publication:

General comments:

  1. Kindly check the entire MS for grammatical errors and English language.

Thanks for your recommendation. We used the extensive English revision, provided by the editing services of the Journal.

  1. What was only unprocessed or minimally processed foods taken as the part of the study?

We complemented lines 102-109 with the following information:

In this ScR, we are interested in studying the presence of HM in the most natural form of the foods, in their raw stages, or the ones with minimally processed methods that are designed to preserve natural foods and could reflect environmental sources of contaminants. Different from natural foods that could be contaminated by intrinsic factors, processed or ultra-processed foods and cooked foods could include chemical contaminants (as additives) or process contaminants (added during the cooking, heating, or storage process of food at home or industries or during their transportation) that have an important impact on food quality and safety and may represent different routes or sources of contamination [14, 20].

  1. It seems a kind of bibliometric analysis therefore; kindly provide a bibliometric graph for the study.

We appreciate your suggestion.

However, we did not analyze data in terms of statistics documents, collaboration index, journal impact, country productivity, document citation analysis, and keywords as it is recommended in bibliometric analysis [4].

Our work is a Scoping review study, so we follow the Joanna Briggs Institute (JBI) methodology for conducting scoping reviews [1] and the PRISMA Extension for Scoping Reviews (PRISMA-ScR) [2]. In this sense the objective of Scoping reviews is to “present a broad overview of the evidence pertaining to a topic, irrespective of study quality, and are useful when examining areas that are emerging, to clarify key concepts and identify gaps” [3]. Therefore, our team considers that a bibliometric graph is not applicable to our methodology.

Specific comments:

Introduction

It is not clear what are the route/sources of HM contamination in these food materials? Kindly supplement this information.

We complemented lines 86-92 with the following information:

HM presence in food varies accordingly to different routes or sources of contamination, defined as intrinsic and extrinsic factors. Intrinsic factors include the season, soil, water, atmospheric deposits, animal feeding regimen [14], and volcanic and vehicle emissions [15]. Also, there are extrinsic factors that contribute to food contamination such as food technological processes, packaging, transportation, and storage [14] culinary procedures tools, and cooking methods [16, 17].

Conclusion

It should be more specific along with future recommendations

Thank you for the comment, we added more specific recommendations to the conclusion in lines 43-47, and to the lines 506-513.

In countries with regulations on these topics, a monitoring system is recommended to evaluate and monitor compliance with national standards. For countries without a regulation system, it is recommended to adopt international guidelines; such as those of FAO, and implement a monitoring system that monitors national compliance. In both cases, the information must be disseminated to the population to create social awareness.

This is especially important both, to protect the population from the consumption of internal production and for the international markets of the globalized world.

Reviewer 2 Report

HM can be accumulated along the food chain, their presence in food is from global concern for human health because some of them are toxic even at low concentrations. This study emphasizes the importance of monitoring the presence of toxic elements in foods and emitting strict regulatory control on food safety in all parts of the world, especially in food producer countries. I recommend it for publication, after some minor clarifications are done.

1. "ppm" is not a official unit, please modify and improve.

2. The clarity of Fig.1 needs to be improved.

3. The source of the map used in this study should be noted. Is the map correct?

4. Please check if the reference format is correct?

Reviewer 3 Report

This review was not written strictly according to the review format and would require major revision before consideration for publication. The specific modifications are as follows:

1.      In this part of Abbreviations, please explain where the content of the article first appears. Please delete it.

2.      Abstract: Please remove these words such as Background, Objective, Methods, etc. In addition, the abstract section of the review should be concise and clear. Please refer to other reviews in this journal for revision.

3.      Bibliography reference format mark, please modify it in full text, such as [1].

4.      There is a problem with the content framework of the entire review, and it cannot be written according to the content of the research article. Please revise it.

Round 2

Reviewer 3 Report

The author has greatly improved on the basis of the original article, so I recommend accepting it in its current form.